# Inoculation and Screening Methods for Major Sorghum Diseases Caused by Fungal Pathogens: *Claviceps africana*, *Colletotrichum sublineola*, *Sporisorium reilianum*, *Peronosclerospora sorghi* and *Macrophomina phaseolina*

**DOI:** 10.3390/plants12091906

**Published:** 2023-05-07

**Authors:** Ezekiel Ahn, Coumba Fall, Jacob Botkin, Shaun Curtin, Louis K. Prom, Clint Magill

**Affiliations:** 1USDA-ARS Plant Science Research Unit, St. Paul, MN 55108, USAshaun.curtin@usda.gov (S.C.); 2Department of Plant Pathology and Microbiology, Texas A&M University, College Station, TX 77843, USA; coumba@tamu.edu; 3Department of Plant Pathology, University of Minnesota, St. Paul, MN 55108, USA; botki009@umn.edu; 4Department of Agronomy and Plant Genetics, University of Minnesota, St. Paul, MN 55108, USA; 5Center for Plant Precision Genomics, University of Minnesota, St. Paul, MN 55108, USA; 6Center for Genome Engineering, University of Minnesota, St. Paul, MN 55108, USA; 7USDA-ARS Southern Plains Agricultural Research Center, College Station, TX 77845, USA; louis.prom@usda.gov

**Keywords:** sorghum, disease evaluation, pathogen inoculation, disease screening, fungal pathogens, *Claviceps africana*, *Colletotrichum sublineola*, *Sporisorium reilianum*, *Peronosclerospora sorghi*, *Macrophomina phaseolina*

## Abstract

Sorghum is the fifth most important crop globally. Researching interactions between sorghum and fungal pathogens is essential to further elucidate plant defense mechanisms to biotic stress, which allows breeders to employ genetic resistance to disease. A variety of creative and useful inoculation and screening methods have been developed by sorghum pathologists to study major fungal diseases. As inoculation and screening methods can be keys for successfully conducting experiments, it is necessary to summarize the techniques developed by this research community. Among many fungal pathogens of sorghum, here we summarize inoculation and screening methods for five important fungal pathogens of sorghum: *Claviceps africana*, *Colletotrichum sublineola*, *Sporisorium reilianum*, *Peronosclerospora sorghi* and *Macrophomina phaseolina*. The methods described within will be useful for researchers who are interested in exploring sorghum-fungal pathogen interactions. Finally, we discuss the latest biotechnologies and methods for studying plant-fungal pathogen interactions and their applicability to sorghum pathology.

## 1. Introduction

Sorghum (*Sorghum bicolor* L. Moench) is a multipurpose food crop that is ranked among the top five cereal crops in the world and is used as a source of food, fodder, livestock feed, and biofuel feedstock [1]. However, sorghum production is significantly constrained by diseases caused by fungal pathogens [2]. Sorghum pathologists have been actively studying the interactions between sorghum and major fungal pathogens, resulting in the development of numerous experimental inoculation and screening techniques. Therefore, it is essential to provide an overview of effective inoculation and screening methods to conduct disease evaluations. Among various fungal pathogens causing sorghum diseases, we summarize the inoculation and screening methods for five major pathogens: *Claviceps africana* Frederickson, Mantle & De Milliano 1991, *Sporisorium reilianum* (Kühn) Langdon & Fullerton, *Peronosclerospora sorghi* (W. Weston & Uppal) C.G. Shaw 1978, *Colletotrichum sublineola* Henn. ex Sacc. & Trotter 1913 (or *sublineolum*) and *Macrophomina phaseolina* (Tassi) Goid 1947. These pathogens were specifically selected due to the importance of diseases caused by the pathogens and our expertise, and Table 1 provides a summary of key information for each of the selected pathogens separately based on their lifecycles. Despite the occasional local outbreak, sorghum ergot disease caused by *C. africana* was not considered an important disease until the introduction of male sterile sorghum in 1960 which rapidly spread worldwide since then becoming one of the major diseases of sorghum [3]. The smut fungus *S. reilianum* causes head smut disease in maize and sorghum, and although the fungus is a biotroph, the disease is devastating for the plant because it leads to complete harvest loss of the affected individual [4]. Sorghum downy mildew, caused by *P sorghi*, is reported in more than 44 countries and can cause severe epidemics, resulting in heavy yield loss [5]. Anthracnose, caused by *C. sublineola*, is one of the most economically damaging diseases of sorghum worldwide, especially in humid and warm areas, and grain yield losses resulting from anthracnose can be over 50% for susceptible lines [6]. *M. phaseolina* is a generalist soil-borne fungus present all over the world that causes diseases such as stem and root rot, charcoal rot, and seedling blight in a wide range of crops such as soybean, sorghum and groundnut [7].

This review provides an overview of the inoculation and screening methods available for the sorghum pathogens, which researchers can use to find suitable methods for future studies. Additionally, we discuss innovative technologies and techniques, such as clustered regularly interspaced short palindrome repeats (CRISPR)/CRISPR-associated protein 9 (Cas9) gene editing, genome-wide association studies (GWAS), and unmanned aerial systems (UAS), which can be applied to the study of sorghum diseases.

## 2. Claviceps africana

### 2.1. Inoculation Methods

Inoculation is a crucial step in the study of plant diseases, and several methods have been developed for this purpose. Tonapi et al. [9] have described three secondary conidia inoculation techniques for sorghum diseases: spray inoculation, brushing technique, and air movement inoculation [9]. For the spray inoculation method, freshly harvested secondary conidia were added to deionized water to make a concentration of 1 × 10^4^ conidia/mL. The suspension was then sprayed onto the panicles using a hand sprayer (Figure 1a). In a similar study, Frederickson and Mantle [10] used diluted honeydew collected from infected panicles and sprayed it on the inflorescences with a hand-operated atomizer. The brushing technique involved collecting secondary conidia from the lids of inverted agar plates with a 1.5 cm wide flat brush. The conidia were then brushed onto wet or dry stigmas (Figure 1b). In comparison, the air movement inoculation technique disseminated secondary conidia grown on a moist soil medium through compressed air using an in-built fan in the wind tunnel (Figure 1c). The sorghum plants with panicles at 50% flowering stage were exposed to the conidia. Panicles were either wet or dry prior to inoculation [9]. According to Tonapi et al. [9], the brushing technique on wet or dry stigmas showed the highest infection rates of 80% and 70%, respectively, while the infection rate for spray inoculation was 32%. The air movement inoculation method had the lowest infection rate of only 3.4% [9].

One simple and effective technique is the dip inoculation method, which involves submerging panicles in a suspension of approximately 1 × 10^6^ conidia/mL [11]. This method is highly effective in inducing infection, as demonstrated in Figure 2a. Another method that has been successfully adapted from the pearl millet inoculation method is the sponge inoculation technique [12], as shown in Figure 2b. This approach involves dipping two 1.25 cm thick sheets of synthetic sponge into a conidial suspension for a few seconds and pressing each panicle between the two sheets of soaked sponge for 2–5 s. When bagged for 7 days post-inoculation, the sponge inoculation method is highly effective, with up to 100% infection rate. These inoculation techniques provide valuable tools for researchers studying *C. africana* and can be adapted to suit specific experimental needs.

### 2.2. Screening Methods

Several studies utilized the percentage of infected spikelets as a screening method, which is determined by dividing the number of infected florets in each panicle by the total number of florets between 7- and 20-days post-inoculation (dpi) [9,11,12,13] In addition, Musabyimana et al. [11] used a 1–5 rating scale where 1 indicated no ergot present, and scores of 4 and 5 were considered susceptible, indicating 26–50% and more than 50% of spikelets were infected by ergot, respectively. Likewise, both the infection rate (%) and the 1–5 rating scale are important screening methods.

### 2.3. Key Facts

Spikelet trimming, a removal of all spikelets that had completed anthesis, ensures inoculation of only non-pollinated and potentially susceptible spikelets [11]. Bagging or misting spray-inoculated panicles with water provides a high humidity favorable for infection and greatly increases the infection rate [11,13]. Spikelets inoculated 1–3 days after anthesis had a negligible ergot infection of 0–2%, which increased progressively from 8.5% in spikelets inoculated 8 h after anthesis to 55.8% in spikelets inoculated 3–7 days before anthesis [11]. Therefore, along with favorable conditions for *C. africana*, it is advantageous to inoculate 3–7 days prior to anthesis to achieve high infection rates.

## 3. Colletotrichum sublineola

### 3.1. Inoculation Methods

Prom et al. [14] outlined the inoculation methods for *C. sublineola* as illustrated in Figure 3a. At the eight to ten-leaf stage, ten sorghum seeds colonized with *C. sublineola* were placed in the leaf whorl of each plant. For spray inoculations, a suspension of 1 × 10^6^ conidia/mL was applied to plants at the growth stage of three (eight-leaf stage) [14]. The two inoculation methods can be used together and have been applied in multiple studies with a few modifications [15,16,17]. The spray inoculation method was also used on Johnsongrass (*Sorghum halepense* [L.] Pers.), a wild relative of sorghum, and sorghum isolates of *C. sublineola* infected Johnsongrass [18]. An excised leaf assay of sorghum to *C. sublineola* was used as a quick screening method for anthracnose (Figure 3b) [19]. This method involved placing a few adaxial leaf pieces of sorghum at the eight-leaf stage on half-strength potato dextrose agar medium (½ PDA), and a droplet of the conidial suspension (1 × 10^6^ conidia/mL) was placed at each side of the excised leaves. The plates were then incubated at 28 °C for four days in the dark [19]. This technique was used on sorghum seedlings [20] and Johnsongrass leaves and rhizomes [21,22] and showed that sorghum seedlings and Johnsongrass were generally highly resistant to *C. sublineola* compared to eight-leaf stage sorghums.

### 3.2. Screening Methods

The traditional method for screening anthracnose used a 1–5 scale system where 0 represented no visible symptoms and 5 represented complete susceptibility [23]. For spray inoculation conducted in the greenhouse and field, disease ratings can also be evaluated based on a 1–5 scale. Prom et al. [14] defined this scale as follows: 1 = no symptoms or chlorotic flecks on leaves, 2 = hypersensitive reaction on the leaves but no acervuli formation, 3 = lesions with acervuli in the centers on the bottom leaves, 4 = necrotic lesions with acervuli on the bottom and middle leaves, and 5 = most leaves dead and infection on the flag leaf containing abundant acervuli. Another study surveyed disease severity on a scale of 1 to 6, where 1 = no symptoms, 2 = 1–5%, 3 = 6–25%, 4 = 26–50%, 5 = 51–75%, and 6 = >75% of leaf area infected [24] The excised-leaf assay developed by Prom et al. [19] evaluated host resistance based on acervuli formation at 4 dpi, where plants were considered susceptible or resistant based on the presence or absence of acervuli. The results of the excised-leaf assay matched with a greenhouse spray inoculation screening method (Figure 4a). Based on the excised-leaf assay, a few modified screening methods have been developed. For example, Ahn et al. [21] scored susceptibility using a 1–5 scale instead of a binary system, where 1 = no infection, 2 = fungal germ tube formed, 3 = fungal bed formed with some imperfectly formed acervuli, 4 = 1–5 acervuli perfectly formed, and 5 = >5 acervuli perfectly formed (Figure 4b). Similarly, Ahn et al. [25] used a 1–10 scale for an excised-leaf assay, and there is a lot of flexibility in screening methods for *C. sublineola*.

### 3.3. Key Facts

Proper temperature and high humidity are essential factors for successful infection by *C. sublineola*, and methods such as bagging or misting can be used to create a favorable environment. Interestingly, some studies have shown that the midrib and leaf blade of sorghum plants respond differently to infection [21,24]. *C. sublineola* can infect sorghum at any stage of growth from seedlings with just a few leaves to mature plants [26]. The inoculation techniques described in this review can be used for sorghum at any growth stage, but it’s important to note that factors like the age of the plants and the experimental location can significantly affect the plant response to the pathogen [15,20].

## 4. Sporisorium reilianum

### 4.1. Inoculation Methods

*Sporisorium reilianum* has been inoculated through various methods over the years. The earliest inoculation method dates to 1910, which involved dusting seeds with smut spores [27]. In greenhouse experiments, Stewart and Reyes [28] found that allowing seedlings to grow to the 4-leaf stage in a spore-soil mixture composed of l/3 spores and 2/3 soil prior to transplanting is an effective method, with an infection rate of 80–100% (Figure 5a). However, this method is not practical for field-based applications [27]. Several other methods with small modifications were attempted, but none were consistently effective in the field [29]. Another inoculation method is by hypodermic injection of inoculum into the apical growing region [27,30]. Mehta [27] compared ten methods of inoculation and found that among them, hypodermic injection showed significantly higher infection rates (Figure 5b). However, this method is not commonly used due to its labor-intensive nature and the difficulty of scaling it up. In 1992, Craig and Frederiksen [31] developed an inoculation method for *S. reilianum* on sorghum seedlings. The method involved infesting the vermiculite surrounding seedling epicotyls with teliospore cultures in 0.25% agar (Figure 5c). The inoculated seedlings were then placed in test tubes containing sterile water deep enough to submerge the first leaf [31,32]. The authors described three different host resistance mechanisms: R1, horizontal resistance to natural infection but susceptibility to all races following syringe inoculation; R2, vertical-specific resistance to some races of *S. reilianum* and susceptibility to others, with the same response to natural infection as to syringe inoculation; and R3, horizontal resistance to natural infection and syringe inoculation [31,32,33]. It is important to note that the outcome of susceptibility tests often differed among various inoculation methods due to different mechanisms of defense [31].

### 4.2. Screening Methods

Plants can be evaluated for *S. reilianum* infection at the heading stage in both greenhouse and field conditions. Plants that exhibit no symptoms have fully developed grains in the main tiller, and no sori are classified as resistant (Figure 6a) [33]. For further verification, plants are evaluated by cutting the main tiller and allowing the side tillers in the ratoon crop to grow to the flowering stage [33]. If disease symptoms are observed in the side tillers, the line is classified as susceptible, while tillers without symptoms are classified as resistant [33]. When using the seedling inoculation method with a spore suspension in agar, susceptible and resistant genotypes can be differentiated based on the presence or absence of brown or dark spots on the first leaf blade within 5 days post-submergence (Figure 6b) [31,32].

### 4.3. Key Facts

The stage of growth is a critical factor in *S. reilianum* infection development [27]. The study from Mehta showed all sorghum plants inoculated at 2–3 weeks old were infected, while infection rates dropped to 92.5% when plants were inoculated at 4 weeks old and only 35% after 5 weeks old [27]. Plants older than 5 weeks at the time of inoculation were completely resistant. Morphological traits including the size, thickness, and number of leaves on the plant also appear to influence infection rates [27]. It is important not to apply water for approximately 5 days after inoculation to facilitate head smut infection [29].

## 5. Peronosclerospora sorghi

### 5.1. Inoculation Methods

Overall, the most effective inoculation methods for promoting downy mildew in sorghum have been compared by Narayana et al. [34], who tested six different techniques (Figure 7). The sandwich inoculation method, previously described by Safeeulla [35], involved placing sprouted seeds in between two infected leaf pieces on a wet Whatman filter paper in a Petri plate, with the abaxial surface facing toward the seeds (Figure 7a). In the spray inoculation method, sprouted seeds on a wet Whatman filter paper were sprayed with a conidial suspension (Figure 7b), while in the dip inoculation, the sprouted seeds were immersed in the conidial suspension for 5 min and the suspension was drained off (Figure 7c). These three methods were followed by incubation for 16 h at 20 °C in the dark, and then sowing in 10 cm-diameter pots in the greenhouse at 25 ± 4 °C with 70–90% relative humidity [34]. For the seedling drop inoculation method, which originated to inoculate pearl millet [36], a micro-syringe was used to place a droplet of inoculum in the whorl of the first leaf stage sorghum seedlings (Figure 7d). The seedlings were also subjected to spray inoculation with a conidial suspension (6 × 10^5^ conidia/mL) (Figure 7e). In the conidial showering method, the outer rim of each pot with seedlings was covered with a layer of moist muslin cloth on which a layer of detached downy mildew-infected leaves was placed with an abaxial surface facing the seedlings (Figure 7f). Two to three layers of blotting paper were placed on top of the infected leaves, and the pots were kept in a tray containing about 1.5 cm of water and covered with another tray lined with moist blotting paper [34]. The pots were incubated overnight at 20 °C to allow the infected leaves to sporulate and the conidia to drop onto the emerging seedlings. According to Narayana et al. [34], seedlings at the one-leaf stage sprayed with a conidial suspension showed the highest systemic infection (100%) in the susceptible lines IS 643 and IS 18433. However, the other five techniques also caused a higher than overall 80% infection rate. Although there are other inoculation techniques and variations, these six methods are generally sufficient for many future studies.

### 5.2. Screening Methods

The susceptibility of sorghum to *P. sorgi* is evaluated at four weeks after planting. Infected plants displaying systemic and/or local lesions are being counted, and the downy mildew incidence rate (%) is calculated. Alternatively, certain thresholds (%) can be set to classify plants as resistant or susceptible for binary evaluation [37].

### 5.3. Key Facts

*P. sorghi*, the causal agent of sorghum downy mildew, is an oomycete and obligate parasite of sorghum. *P. sorghi* can also infect maize but only can complete its lifecycle on sorghum, producing diploid oospores [38]. In the asexual phase of the *P. sorghi* life cycle, sporangiophores develop from lesions with sporangia. Being an obligate pathogen, the inoculum is produced *in planta*. Isolates are collected from the field in the form of oospore-infested leaf tissue and maintained on infected plants. Sporulation of the pathogen is optimal at 22 °C and does not occur above 26 °C or below 10 °C after an 8-h incubation period [38]. Deposition of sporangia is highest 8 h after incubation starts and ceases after 10 h [38]. Systemically infected leaves release large amounts of sporangia, which can be used to prepare conidial suspensions.

## 6. Macrophomina phaseolina

### 6.1. Inoculation Methods

Field and greenhouse methods have been described to screen the response of sorghum genotypes to *M. phaseolina*, the causal agent of sorghum charcoal rot. Field screening techniques include the sick-plot (Figure 8) and toothpick (Figure 9) inoculation methods described by Das et al. [39], while greenhouse procedures are described by Das et al. [39] and Bandara et al. [40] (Figure 10 and Figure 11).

The sick-plot method is a field screening technique for evaluating the response of sorghum genotypes to *M. phaseolina*. Genotypes of the same maturity group are grown in replicated trials with an inoculum density of 100–150 microsclerotia per gram of soil (Figure 8a). Plants are spaced uniformly, and a susceptible check is included for every twenty plants within each replicate (Figure 8a). A resistant check is also inserted in the trial. Homogeneous moisture stress is induced at the onset of flowers by either ceasing to supply water or suppressing the flag leaf or both (Figure 8b). An overview of the procedure is shown in Figure 8.

The toothpick method involves submerging cleaned and dried bundles of wooden toothpicks in Potato Dextrose Broth (PDB) and sterilizing them (Figure 9a,b). The toothpicks are then inoculated with an actively growing culture of *M. phaseolina* and incubated for ten days at 25 ± 1 °C (Figure 9c). The next step is to conduct a replicated field trial, where an infested toothpick is inserted into the second internode of 5–10 plants per row in each replicate, 10–50 days after 50 % flowering (Figure 9d). The inoculated plants should be uniform in growth and spacing, and moisture stress should be induced at 50% flowering. Figure 9 provides an overview of the protocol for screening sorghum genotypes for their response to *M. phaseolina* using the toothpick method.

In phaseolinone sensitivity evaluation, seedling tissues of the tested genotypes are evaluated for their sensitivity to phaseolinone, a predominant exotoxin produced by *M. phaseolina* in cell-free culture filtrate (CFCF). Known concentrations of CFCF or purified phaseolinone are used to test 15-day-old seedlings of the tested genotypes, along with resistant and susceptible checks. CFCF is obtained and adapted from the method described by Saha et al. [41]. An Erlenmeyer flask containing sterilized PDB is inoculated with plugs of culture media colonized by *M. phaseolina* (Figure 10a) and shaken for 8–9 days at 150 rpm and 25–30 °C (Figure 10b). The CFCF is then separated from the mycelium by filtration using a Whatman filter paper No.1 (Figure 10c). The harvested filtrate can then either be used to directly test the genotypes or serve as a substrate to purify the phaseolinone following the method described by Zheng et al. [42] (Figure 10d). Thus, known concentrations of raw CFCF or purified phaseolinone, resulting from a serial dilution (Figure 10e) are used to inoculate the seedlings (Figure 10f). The symptoms are evaluated ten days after inoculation (Figure 10g). Figure 10 summarizes the steps followed for this method.

To evaluate charcoal rot in a greenhouse setting, *M. phaseolina* is first grown on PDA at 30 °C for five days. Colonized PDA plugs (3–4 mm^2^) are then transferred to a 500 mL Erlenmeyer flask containing 100 mL PDB and shaken for five days at 60 rpm and room temperature under continuous light. The resulting suspension is blended at 18,000 rpm for 10 min using a Waring blender (Figure 11a) and then filtered through four layers of sterile cheesecloth to obtain small fragments (Figure 11b). The filtrate is then centrifuged at 3000× *g* for 5 min to obtain a pellet, which is resuspended in 50 mL of 10 mM sterile phosphate-buffered saline (PBS) (pH = 7.2). The concentration is adjusted to 5 × 10^4^ hyphal fragments per milliliter by dilution using PBS (Figure 11d), and the viable fragments are determined by plating serially diluted suspensions onto PDA (Figure 11e). The colony-forming units are counted for 10 replicate PDA plates after three days of incubation at 30 °C, and the mean is calculated (Figure 11f,g). Inoculation is performed 14 days after flowering. Three plants of each tested genotype are injected with approximately one milliliter of inoculum into the basal node of the stalk using a custom-made Hamilton metal 15-gauged 51 mm hub needle (Fisher Scientific, Pittsburgh, PA, USA) (Figure 11h). The negative control is injected with sterile PBS.

### 6.2. Screening Methods

The Sick-plot Method involves recording plant stand and height from the beginning of growth until 50% flowering (Figure 8c). At harvest time, various measurements are taken, including (a) the number of plants affected by charcoal rot and lodging, (b) the length of lesions, (c) the number of nodes impacted by lesions, and (d) grain yield (Figure 8d,e). Disease incidence and severity are then calculated (Figure 8f), using the disease severity scale developed by Das et al. [39] (Table 2). By comparing the disease incidence and severity of tested genotypes with those of resistant and susceptible checks, potential sources of resistance can be identified.

Using the toothpick method, each stalk is examined for charcoal rot symptoms between 25–35 days after inoculation (Figure 9e,f). Disease incidence and severity are then calculated, following the same approach as the sick-plot method. The resulting data is used to compare the tested genotypes with the resistant and susceptible checks to identify potential sources of resistance.

To evaluate resistance to phaseolinone sensitivity, susceptible genotypes are expected to display symptoms and eventually die. The resistance of the tested genotypes can be assessed by comparing them to the resistant checks, using the same approach as described in the sick-plot screening methods. The resistant phenotypes can be further validated by conducting field tests using the sick-plot procedure to confirm the accuracy of the results obtained.

The greenhouse charcoal rot screening method involves recording various parameters 56 days after inoculation, which is approximately 70 days after flowering. These parameters include measuring the plant height from the soil surface to the tip of the panicle, determining the stalk diameter and lesion length, counting the number of nodes crossed by the lesion, and measuring the 100 seed weight, total seed weight per panicle, and a total number of seeds per panicle. These measurements are taken to evaluate the potential impact of charcoal rot on the tested genotypes and to compare them to resistant and susceptible checks. By analyzing these data points, the researchers can identify sources of resistance and develop effective strategies for managing this disease.

### 6.3. Key Facts

To optimize the sick-plot method for screening potential resistance to charcoal rot, the susceptible check should develop at least 50% disease incidence. However, non-uniform inoculation may lead to potential escapes, which should not be confused with resistance. The toothpick screening method, on the other hand, is useful for studying variability among *M. phaseolina* isolates and can be modified to detect different levels of resistance in tested genotypes. Additionally, phaseolinone sensitivity assays have been suggested for quickly screening large germplasms, though confirmation in the field is necessary [39]. Another method for evaluating charcoal rot resistance is the greenhouse evaluation method described by Bandara et al. [40]. This method employs the resistance-tolerance index (Index*_RT_*), where lower values indicate greater disease resistance and higher tolerance. The authors provide a detailed procedure for this method in their publication and demonstrate that Index*_RT_* is more robust than the commonly used lesion length in predicting yield losses.

## 7. Innovative Biotechnologies and Methods for Detecting and Evaluating Diseases in Sorghum

Recent advancements in machine learning and computer vision technologies have opened new possibilities for high-throughput plant phenotyping by using UAS [43]. In a study, a convolutional neural network (CNN) model was used to detect morphological traits in sorghum demonstrating the effectiveness of this approach [43]. The effectiveness of Convolutional Neural Networks in image recognition motivates researchers to extend its applications in the field of agriculture for the recognition of plant species, yield management, weed detection, soil, and water management, fruit counting, diseases, and pest detection, evaluating the nutrient status of plants, and much more [44]. In another study, the authors proposed an innovative sorghum leaf disease detection with a dataset of 260 images and a classification method using a convolutional neural network [45]. The application of deep learning is expected to increase the accuracy and speed of disease detection and evaluation. Furthermore, hyperspectral imaging provides a novel 3D deep convolutional neural network (DCNN) that directly assimilates the hyperspectral data in plant disease detection [46]. Although not fungal diseases, Wang et al. [47] developed a CRISPR/Cas12a-based visual nucleic acid detection system targeting sorghum mosaic virus and rice stripe mosaic virus. In the future, biotechnologies could revolutionize fungal pathogen detection using CRISPR/Cas technology, as well as provide insights into the gene expression of both plants and pathogens, potentially leading to more effective disease control and treatment strategies.

Most sorghum diseases are relying on novel sources of resistant genes, and GWAS, a test of hundreds of thousands of genetic variants across many genomes to find those statistically associated with a specific trait or disease [48] has been more successful in plants than in humans [49]. Recent studies revealed potential candidate genes of sorghum that confer resistance against various fungal pathogens including anthracnose, head smut and downy mildew [50], but validation of candidate genes possibly conferring for the traits can be made. CRISPR/Cas9 mediated targeted mutagenesis in sorghum has been applied to target genes with various traits [51,52]. CRISPR enzymes require a protospacer-adjacent motif (PAM) near the target cleavage site, constraining the sequences accessible for editing, and recently engineered Cas9-Sc++ and a higher-fidelity mutant HiFi-Sc++ extend the use of CRISPR editing for diverse applications [53]. Rapid Quantitative Evaluation of CRISPR Genome Editing by TIDE (Tracking of Indels by Decomposition) and TIDER (Tracking of Insertions, DEletions, and Recombination events) can accurately identify and quantify insertions and deletions [54]. Recent developments of Prime editors (PEs), which can install desired base edits without donor DNA or double-strand breaks, have been used in plants and can, in principle, accelerate crop improvement and breeding [55]. Finally, Yang et al. [56] discovered a heritable transgene-free genome editing technique in plants by grafting wild-type shoots to transgenic donor rootstocks. Once the system is established in sorghum, it could potentially accelerate the validation of gene functions against fungal pathogens.

## 8. Conclusions

Various inoculation and screening methods have been developed for the pathogens discussed in this review, and many of them can be applied to other plants including crops and weeds as pathogen inoculation and screening methods have general commonalities. For instance, multiple inoculation methods for *C. sublineola* and *S. reilianum* have been tested on Johnsongrass for anthracnose and head smut resistance, respectively. While *C. sublineola* successfully colonized Johnsongrass [18,21], *S. reilianum* inoculation using the method described by Craig and Frederickson did not show any signs of infection [31,32]. This may suggest that Johnsongrass is resistant to *S. reilianum* (either universally or specific pathotype(s)), or other inoculation methods may be more suitable. Similarly, *C. africana*, *P. sorghi* and *M. phaseolina* are reported to infect Johnson grass [57], but not many studies have been reported for detailed cross-infections between sorghum and Johnsongrass regarding the pathogens. Therefore, it is crucial to test classic and modern inoculation and screening methods while continuously developing novel methods of inoculation and screening for these pathogens to ensure that the outcomes can be consistent and reliable. The availability of cutting-edge technologies with deep learning for studying plant pathology suggests the possibility of novel accurate and rapid disease evaluation methods in sorghum in the future, and by combining currently available methods, we are expecting great innovations in the field [58].

## Figures and Tables

**Figure 1 plants-12-01906-f001:**
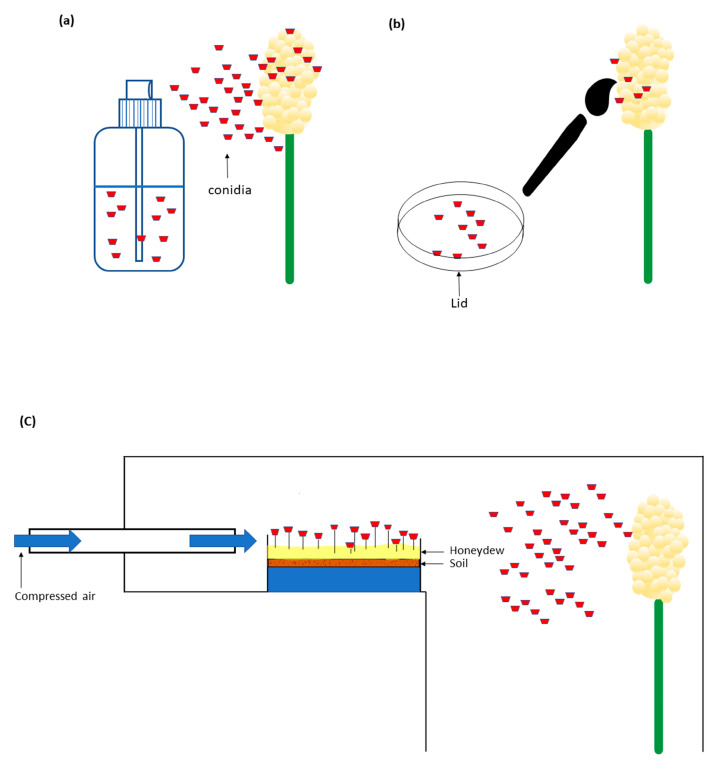
Illustrations of the three secondary conidia inoculation techniques described by Tonapi et al. [9]. (**a**) Spray inoculation of secondary conidia (concentration: 1 × 10^4^ conidia/mL). (**b**) Stigmas (wet or dry) were brushed with secondary conidia collected from the lids of inverted agar plates. (**c**) Secondary conidia were applied using compressed air.

**Figure 2 plants-12-01906-f002:**
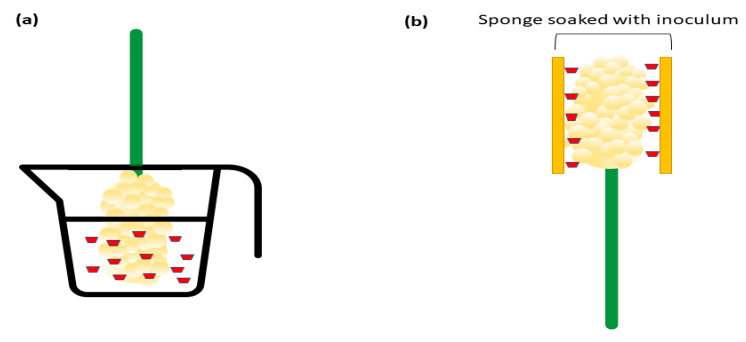
Illustrations of the dip inoculation and sponge inoculation methods. (**a**) Panicles were dipped into inoculum with approximately 1 × 10^6^ conidia/mL [11]. (**b**) A sponge soaked with *C. africana* spore suspension is physically in contact with panicles [12].

**Figure 3 plants-12-01906-f003:**
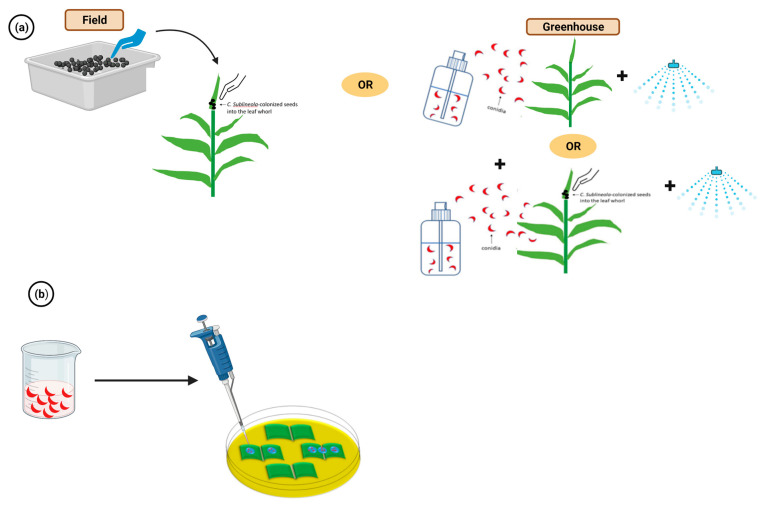
Illustrations of the spray inoculation and excised-leaf inoculation methods (**a**) 1 × 10^6^ conidia/mL of inoculum was sprayed and colonized sorghum seeds were dropped into the leaf whorl at 8 leaf stage sorghum plants [14]. (**b**) Excised leaves on ½ PDA plates were droplet inoculated with 1 × 10^6^ conidia/mL of inoculum [19]. Compared to the typical spray inoculation method, the excised leaf assay shortens the waiting time for screening as the leaves can be scored at 4 dpi. Created with BioRender.com (accessed on 11 January 2023).

**Figure 4 plants-12-01906-f004:**
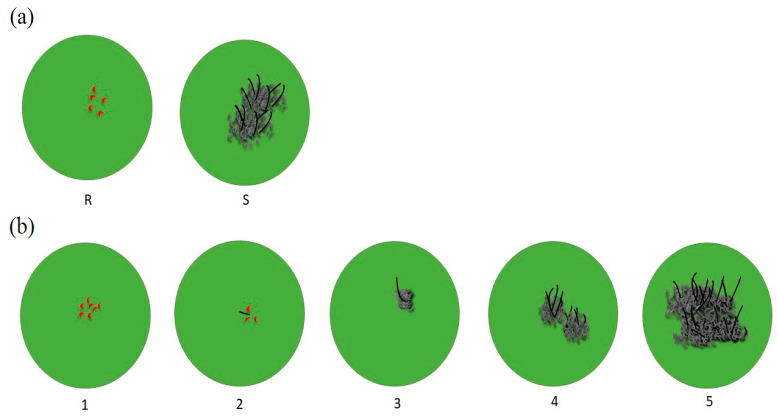
Screening methods for an excised-leaf assay. (**a**) Typical screening method based on the presence or absence of acervuli formation at 4 dpi [19]. This binary system is useful for quick screening of host resistance, where plants are considered susceptible if acervuli are present and resistant if acervuli are absent. (**b**) A modified screening method using a 1–5 scale [21]. This system provides more resolution for scoring the severity of infection. The scale ranges from 1 (no infection) to 5 (>5 acervuli perfectly formed), with intermediate scores for different levels of infection.

**Figure 5 plants-12-01906-f005:**
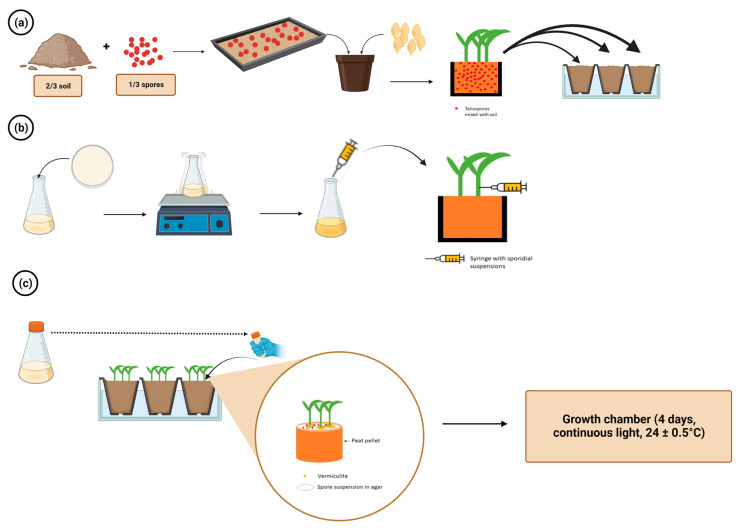
Three different inoculation methods of *S. reilianum* to sorghum. (**a**) Teliospores mixed with soil followed by planting seeds, (**b**) Hypodermic injection of sporidial suspension to sorghum seedlings and (**c**) Seedling inoculation with spore suspension in agar.

**Figure 6 plants-12-01906-f006:**
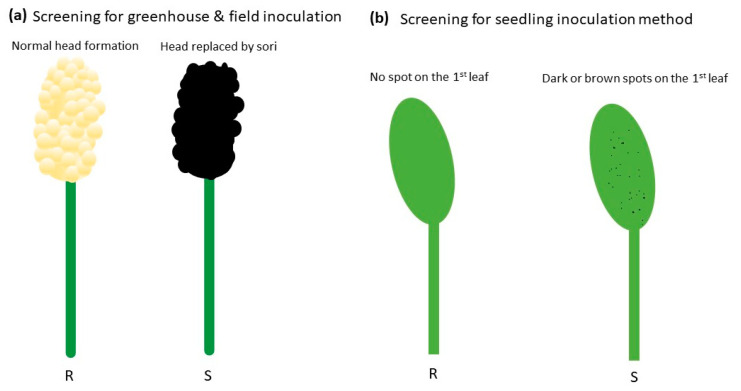
Illustrations show screening methods for *S. reilianum* in greenhouse & field and excised-leaf assays. (**a**) Screening is based on the presence or absence of infected heads in plants with fully developed grains and no sori. (**b**) Screening for the seedling agar inoculation method is based on the presence or absence of dark or brown spots on the first leaf.

**Figure 7 plants-12-01906-f007:**
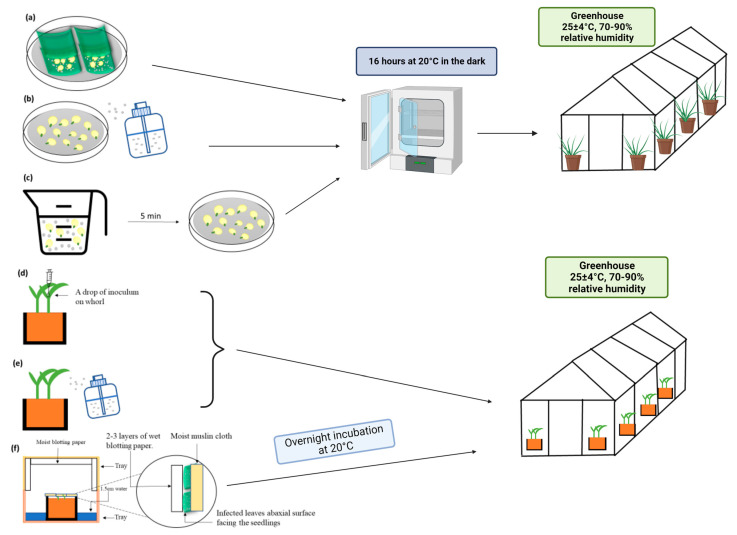
Six *P. sorghi* inoculation methods for sorghum described by Narayana et al. [34]. (**a**) Sandwich inoculation of sprouted seeds placed between two infected leaf pieces on a wet Whatman filter paper in a Petri plate. (**b**) Spray inoculation of sprouted seeds with a conidial suspension. (**c**) Dip inoculation of sprouted seeds immersed in a conidial suspension. (**d**) Drop-inoculated seedlings at the one-leaf stage using a micro-syringe to place a droplet of inoculum in the whorl of the first leaf. (**e**) Spray inoculation of seedlings at the one-leaf stage with a conidial suspension. (**f**) Conidial showering of seedlings by covering the outer rim of each pot with moist muslin cloth and a layer of detached downy mildew-infected leaves, allowing the conidia to drop onto the emerging seedlings.

**Figure 8 plants-12-01906-f008:**
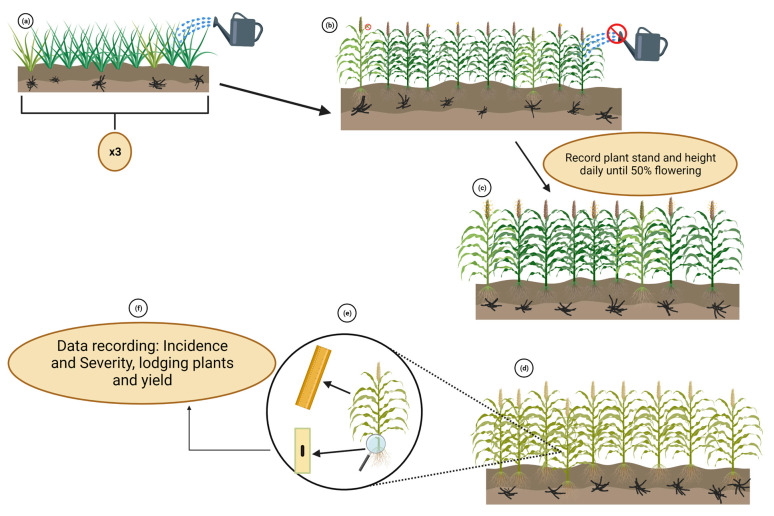
Overview of the sick-plot method. Black masses in soil represent the microsclerotia of *M. phaseolina*. (**a**) tested genotypes in green, light green corresponds to susceptible checks. (**b**) Moisture stress induction. (**c**–**f**) Data recording for screening purposes. Created with BioRender.com (accessed on 11 January 2023).

**Figure 9 plants-12-01906-f009:**
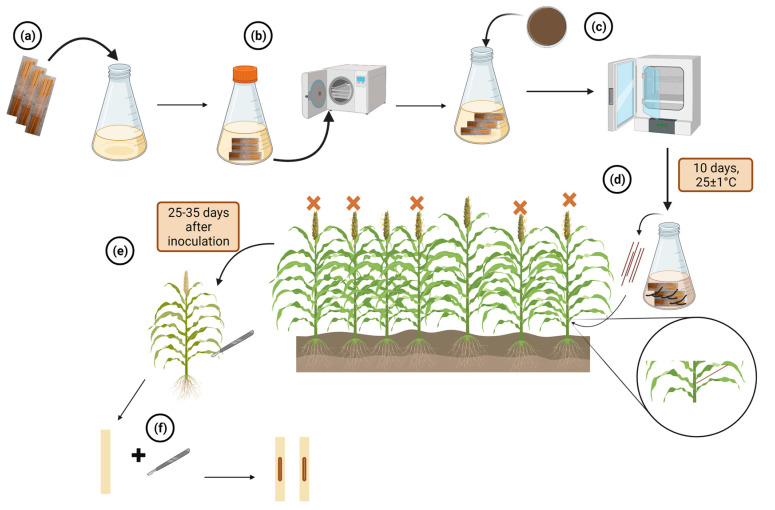
Illustrations for the toothpick method. (**a**) Wooden toothpicks are submerged in PDB. (**b**) Wooden toothpicks in PDB are sterilized. (**c**) Toothpicks are inoculated with *M. phaseolina* and incubated for ten days at 25 ± 1 °C. (**d**) Infested toothpicks are placed into the second internode. (**e**,**f**) Sampling and evaluating inoculated plants. 4 Orange crosses indicate inoculated plants. Created with BioRender.com (accessed on 11 January 2023).

**Figure 10 plants-12-01906-f010:**
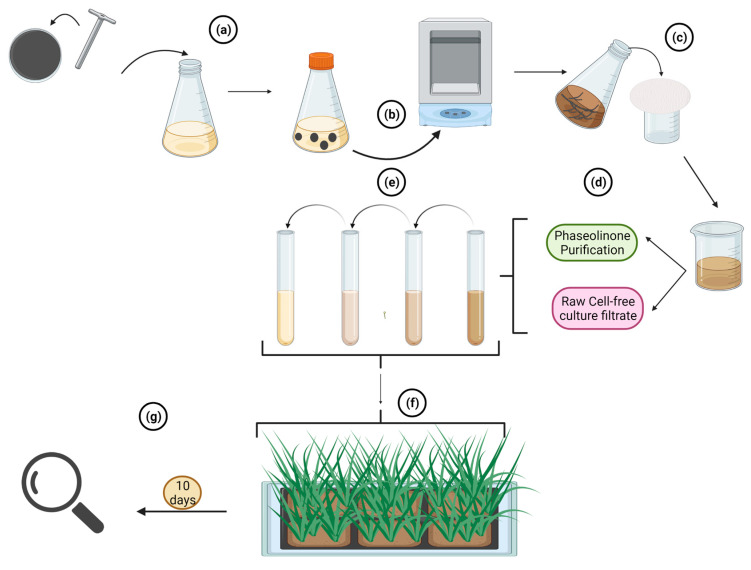
The Phaseolinone sensitivity screening procedure. (**a**) The procedure starts with the inoculation of sterilized PDB with *M. phaseolina* in a capped Erlenmeyer flask (**b**) The culture is then shaken for 8–9 days at 150 rpm and 25–30 °C. (**c**) Afterward, the culture is filtered through a Whatman filter paper No.1 to harvest the filtrate. (**d**) The harvested filtrate is directly used to test the genotypes or serve as a substrate to purify the phaseolinone. (**e**) A serial dilution of the culture is conducted to prepare known concentrations of the raw CFCF or purified phaseolinone. (**f**) The seedlings of the tested genotypes are then inoculated with the prepared dilutions of *M. phaseolina*. (**g**) Evaluations for diseases are recorded at 10 dpi. Created with BioRender.com (accessed on 11 January 2023).

**Figure 11 plants-12-01906-f011:**
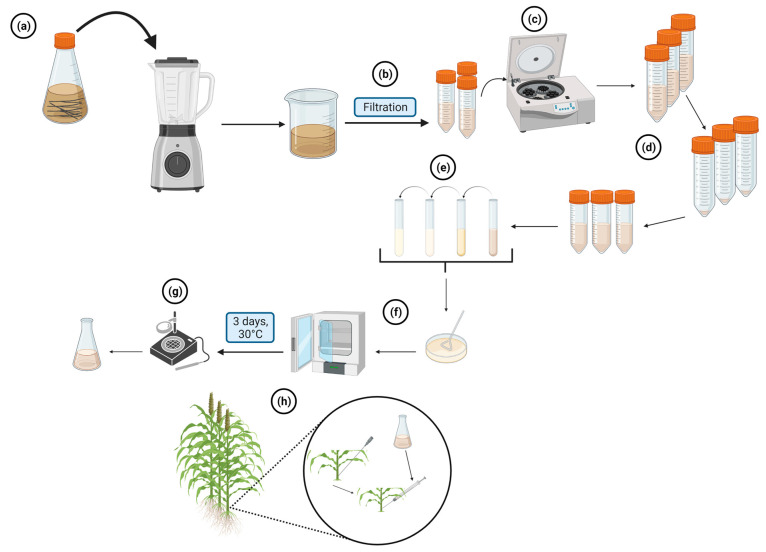
Inoculum preparation for charcoal rot greenhouse evaluation. (**a**) *M. phaseolina* is grown on PDA and then blended at high speed for 10 min using a Waring blender. (**b**) The suspension is filtered through four layers of sterile cheesecloth to obtain small fragments. (**c**) The resulting suspension is centrifuged, and the pellet is resuspended in 50 mL of PBS. (**d**) The concentration is adjusted to 5 × 10^4^ hyphal fragments per milliliter by diluting the suspension using PBS. (**e**) The suspension goes through multiple steps of dilution and is spread onto a PDA plate. (**f**) The PDA plates are incubated at 30 °C for three days, and (**g**) the colony-forming units are counted for ten replicate plates. (**h**) Inoculation is performed on sorghum plants 14 days after flowering by injection. Created with BioRender.com (accessed on 11 January 2023).

**Table 1 plants-12-01906-t001:** Overview of fungal diseases of sorghum discussed in the review [8].

Pathogen	Disease	Phylum	Lifestyle	Overwintering Structure	Primary Inoculum	Secondary Inoculum	Environmental Conditions
*Claviceps africana*	ergot	Ascomycota	Biotrophic	Unclear, but potentially sclerotia	Unclear, but potentially sclerotia	Primary and secondary conidia from honeydew	Cool and rainy weather at anthesis
*Sporisorium reilianum*	Head smut	Basidiomycota	Biotrophic	Teliospore	Teliospore	Soilborne Teliospore	Favorable environmental conditions are poorly understood
*Peronosclerospora sorghi*	Downy mildew	Oomycota	Biotrophic	Oospore	Oospore	Conidia	Minimum soil temperature at 10 °C Conidial production at 18 °C
*Colletotrichum sublineola*	Anthracnose	Ascomycota	Hemibiotrophic	Microsclerotia	Microsclerotia Seed transmission Alternative hosts (ex: Johnsongrass)	Conidia	Light at 22–30 °C
*Macrophomina phaseolina*	Charcoal rot	Ascomycota	Hemibiotrophic	Sclerotia	Sclerotia	Conidia	Hot and dry weather conditions High soil temperature (35–37 °C)

**Table 2 plants-12-01906-t002:** Disease Severity Rating Scale for Charcoal Rot in Sorghum: The Sick-plot Method [39].

Disease Rating	Description
1	One internode invaded but the rot does not pass through any nodal area.
2	Two internodes invaded.
3	Three internodes invaded.
4	More than three internodes invaded.
5	Five most internodes invaded with the shredding of stalks and death of plants.

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
