# Peer review of "Inoculation and Screening Methods for Major Sorghum Diseases Caused by Fungal Pathogens: Claviceps africana, Colletotrichum sublineola, Sporisorium reilianum, Peronosclerospora sorghi and Macrophomina phaseolina"

_plants, 2023, doi:10.3390/plants12091906_

Round 1
Reviewer 1 Report
I think that this paper provides transparent review about the main inoculation and screening methods for major sorghum disease caused by selected fungal pathogen and will serve as a useful tool for researchers and breeders. However, I think that some reorganization of the manuscript text and addition of some important information is need for better understanding, clarification the text and increasing its scientific value. I propose this changes and addition at manuscript:
The main proposed changes and addition:
1) Write summarizing paragraph about pathogen at the beginning of each discussed disease. Information mentioned at Table 1 seems me not sufficient.
2) Add information about usage mention methods to other diseases to each mentioned inoculation and screening method at each pathogen. It is enough only shortly, but it is missing there. At paragraph 8 conlusion, you mention it only according to pathogen C. sublineola. However, to mention it there at general conclusion it is not optimal. It would be more suitable to include it at the text discussing individual pathogen. At key facts at each pathogen at each method, there should be discussed only advantages or disadvantages of individual method.
3) Text about Macrophomina phaseolina inoculation and screening methods is wrote more in details comparing the others. It is needed to unify it at the text.
4) At pathogen Macrophomina phaseolina, phaseoline sensitivity evaluation (on page 13) move from inoculation to screening method.
The others:
1) Do some changes at Table 1. It provides the basic information about discussed diseases and comparison among them. It fulfill this purpose well. I propose delete the last column. The table is spreading and to mention the citation at the end of table title …according to (3)
Order pathogen at the table according some aspect. It is not clear if it is according to the worldwide distribution on sorghum or anything else. I propose sorted them by lifestyle or phylum. Firstly, discuss biotrophic pathogen then hemibiotrophic. Put text together it. However it is up to you but it is needed to clarify it at the text.
2) At paragraph 8 conclusion, I disagree partially about some statement you wrote there on line 462… It is crucial to develop novel methods of inoculation and screening… Yes, it is important to develop new molecular methods but the crucial is to know the main (old or new) inoculation and screening method for individual pathogen, theirs advantages or disadvantages. According that you should choose the most suitable method correctly. I propose to rewrite paragraph 8. Conclusion to be generally valid.
Author Response
1) Write summarizing paragraph about pathogen at the beginning of each discussed disease. Information mentioned at Table 1 seems me not sufficient. - The correction was made accordingly.
2) Add information about usage mention methods to other diseases to each mentioned inoculation and screening method at each pathogen. It is enough only shortly, but it is missing there.- We agree that this will help out improving the manuscript, but it is very hard to track down studies that used the methods introduced because many classic inoculation methods are from extremely old literature (the 1950s or before).
At paragraph 8 conlusion, you mention it only according to pathogen C. sublineola. However, to mention it there at general conclusion it is not optimal. It would be more suitable to include it at the text discussing individual pathogen. - The correction was made accordingly.
At key facts at each pathogen at each method, there should be discussed only advantages or disadvantages of individual method. - Although we can speculate, it is hard to tell what are advantages or disadvantages of individual methods without direct comparisons.
3) Text about Macrophomina phaseolina inoculation and screening methods is wrote more in details comparing the others. It is needed to unify it at the text. - We are aware of the problem, but it is the nature of the pathogen and the methods are more complicated than others.
4) At pathogen Macrophomina phaseolina, phaseoline sensitivity evaluation (on page 13) move from inoculation to screening method.- It mentions that the next step is screening, but it doesn't describe the actual screening method.
The others:
1) Do some changes at Table 1. It provides the basic information about discussed diseases and comparison among them. It fulfill this purpose well. I propose delete the last column. The table is spreading and to mention the citation at the end of table title …according to (3) - The correction was made accordingly by deleting the last column and fixing the citation.
Order pathogen at the table according some aspect. It is not clear if it is according to the worldwide distribution oe. I propon sorghum or anything elsse sorted them by lifestyle or phylum. Firstly, discuss biotrophic pathogen then hemibiotrophic. Put text together it. However it is up to you but it is needed to clarify it at the text. - The correction was made accordingly.
2) At paragraph 8 conclusion, I disagree partially about some statement you wrote there on line 462… It is crucial to develop novel methods of inoculation and screening… Yes, it is important to develop new molecular methods but the crucial is to know the main (old or new) inoculation and screening method for individual pathogen, theirs advantages or disadvantages. According that you should choose the most suitable method correctly. I propose to rewrite paragraph 8. Conclusion to be generally valid.- The correction was made accordingly by modifying paragraph 8.
Reviewer 2 Report
It would be good to add original pictures at least of the symptoms produced on experimental plants.
Author Response
It would be good to add original pictures at least of the symptoms produced on experimental plants.- We are aware of the problem, but we currently don't have good-quality pictures of the symptoms caused by the diseases that don't violate copyright issues.
Reviewer 3 Report
MS provides a thorough and comprehensive overview of the inoculation and screening methods of the most important fungal pathogens of millet. Neither the previous century's nor the modern molecular biology descriptions are left out. The manuscript is richly illustrated, but at the same time the first table is still formally unorganized and the inscriptions on several figures are unreadably small (Fig. 3, fig. 5, fig. 7).
A few minor editing errors:
ad 460: extra space (?)
ad 528: The highlighting of the Latin name has been omitted
Congratulations on this outstanding quality work!
Author Response
The manuscript is richly illustrated, but at the same time the first table is still formally unorganized and the inscriptions on several figures are unreadably small (Fig. 3, fig. 5, fig. 7)- We are aware of the problem and high-resolution pictures will be submitted for the figures.
A few minor editing errors:
ad 460: extra space (?)- The correction was made accordingly
ad 528: The highlighting of the Latin name has been omitted- The correction was made accordingly
Congratulations on this outstanding quality work!- We appreciate your thorough review.